# Design, Synthesis and Bioactivity Evaluation of 4,6-Disubstituted Pyrido[3,2-*d*]pyrimidine Derivatives as Mnk and HDAC Inhibitors

**DOI:** 10.3390/molecules25184318

**Published:** 2020-09-21

**Authors:** Kun Xing, Jian Zhang, Yu Han, Tong Tong, Dan Liu, Linxiang Zhao

**Affiliations:** Key Laboratory of Structure-Based Drug Design & Discovery of Ministry of Education, Shenyang Pharmaceutical University, Shenyang 110016, China; 15041472352@163.com (K.X.); ZJ18241498987@163.com (J.Z.); hanyu19940211@163.com (Y.H.); tong20141109@163.com (T.T.)

**Keywords:** HDAC and Mnk inhibitor, pyrido[3,2-*d*]pyrimidine derivatives, antiproliferative activity

## Abstract

Both HDACs and Mnks play important role in translating multiple oncogenic signaling pathways during oncogenesis. As HDAC and Mnk are highly expressed in a variety of tumors; thus simultaneous inhibit HDAC and Mnk can increase the inhibition of tumor cell proliferation and provide a new way of inhibiting tumor growth. Based on the previous work and the merge pharmacophore method; we designed and synthesized a series of 4,6-disubstituted pyrido[3,2-*d*]pyrimidine derivatives as HDAC and Mnk dual inhibitors. Among them; compound **A12** displayed good HDAC and Mnk inhibitory activity. In vitro antiproliferative assay; compound **A12** exhibited the best antiproliferative activity against human prostate cancer PC-3 cells. Docking study revealed that the pyrido[3,2-*d*]pyrimidine framework and hydroxamic acid motif of compound **A12** were essential for maintaining the activity of HDAC and Mnk. These result indicated that **A12** was a potent Mnk /HDAC inhibitor and will be further researched.

## 1. Introduction

Histone deacetylase (HDAC) plays an important role in regulating the acetylation level of histones or non-histone proteins, thus it can regulate cell proliferation, differentiation and apoptosis [1]. The abnormal changes in histone acetylation levels often induce the expression of oncogene. Thus, targeting HDAC could be a strategy to treat cancer. HDAC inhibitors can inhibit cell growth and differentiation by increasing histone and non-histone acetylation levels in tumor cells, and further promote apoptosis. So far, five HDAC inhibitors have been approved by the U.S. FDA and a large number of HDAC inhibitors have been reported. Although HDAC inhibitors show promising therapeutic effects in hematological cancers, but they show poor effect in solid tumors. For example, vorinostat has an unsatisfactory effect on breast cancer, colorectal cancer, non-small cell lung cancer and thyroid cancer [2]. Romidepsin, belisestat and pabistil are also not effective in solid tumors [3]. On the other hand, HDAC inhibitors have distinct side effects, which affect the patients’ medication compliance, so simultaneously inhibiting HDAC and another target could show a higher therapeutic effect than only inhibiting HDAC.

Multitarget-directed ligands are agents that treat multifactorial disease by interacting with multiple target involved in pathogenesis [4]. This strategy can avoid many pitfalls, such as dose-limiting toxicities, drug−drug interactions and complex and unpredictable pharmacokinetic profiles. Dual/multi-target inhibitors based on HDAC and kinases have made significant progress in the last decade [5]. Among them, CUDC-101 [6] and CUDC-907 [7] were already in clinical research stage (Figure 1). CUDC-101, which was obtained by combining vorinostat and erlotinib, could simultaneously inhibit HDAC, EGFR, HER2, whose IC_50_ value against HDAC, EGFR and HER2 were 4.4 nM, 2.4 nM and 15.7 nM, respectively. In multiple tumor cell lines, CUDC-101 has better antiproliferative effect than vorinostat, erlotinib and the combination of vorinostat and erlotinib. Currently, CUDC-101 is in phase I clinical trials. CUDC-907, which could simultaneously inhibit HDAC and PI3K, inhibited tumor growth in a dose-dependent manner in a Daudi non-Hodgkin lymphoma xenograft model. In vivo anti-tumor activity studies have shown that CUDC-907 can cause accumulation of Ac-H3 and a decrease in p-AKT levels. CUDC-907 is currently undergoing a phase II clinical trial for patients with lymphoma and advanced thyroid carcinoma. Although a large number of multiple inhibitors based on inhibit HDAC and other targets have been reported, only two compounds have been in clinical research. Therefore, it is necessary to develop multi-target inhibitors based on HDAC.

Mitogen-activated protein kinase interacting kinases (Mnk1 and Mnk2) belong to serine/threonine kinase and are the only known kinases that can phosphorylate eIF4E at Ser209 [8,9]. In humans, Mnks are presented in two isoforms: Mnk1 and Mnk2, and they share 80% sequence identity [10]. So it is feasible to simultaneously inhibite Mnk1 and Mnk2. Although the MnK signaling pathway is branched at multiple directions, but the Mnk double knock-out does not affect the normal growth of mice [11], which indicated that Mnks are dispensable in normal cells. In addition, Mnks can mediate the resistance of many clinical drugs [12,13], such as Imatinib and Cytarabine. Therefore, the combination of Mnk inhibitors or the design of dual-target inhibitors that simultaneously targeting Mnk and other targets may have better anti-tumor effects. In summary, inhibiting Mnks is expected to become a low-toxic and efficient anti-tumor solution.

So far, no Mnk inhibitor has been approved for marketing. Only three Mnk inhibitors have entered clinical research (Figure 1). BAY1143269 [14], which is a selective Mnk1 inhibitor and which structure has not been disclosed, was combined with docetaxel in a clinical phase I study for the treatment of non-small cell lung cancer, but that clinical study has now been terminated [15]. eFT508 is a selective Mnk inhibitor that has entered phase II clinical research for the treatment of solid tumors [16]. ETC-206 combined with dasatinib entered a clinical phase I study for the treatment of blast crisis-chronic myeloid leukemia [17]. We previously found that 4,6-disubstituted pyrido[3,2-*d*]pyrimidine derivatives as good Mnk inhibitors [18]. Among them, compound **D06** showed good Mnk inhibitory activity, the IC_50_ against Mnk1 and Mnk2 were 1.1 nM and 26 nM, respectively.

Mnk is located downstream of the MAPK signaling pathway and thus can regulate cell survival and proliferation. What’s more, HDAC and Mnk are highly expressed in a variety of tumors, and there are cases where both are overexpressed in a variety of cancers, such as colon cancer, prostate cancer and leukemia. In addition, HDAC and Mnk regulate the expression of some common oncoproteins, such as c-Myc, Cyclin D1 and VEGF (Figure 2). Therefore, small molecule inhibitors that target both HDAC and Mnk can increase the inhibition of tumor cell proliferation and provide a new way of inhibiting tumor growth.

Molecular docking performed between **D06** and Mnk2 (PDB code 2hw7) showed that the pyrido[3,2-*d*]pyrimidine core could imitate the structure of purine in ATP and made hydrogen bond interaction with Met162 in hinge. The 4-fluoroaniline group could occupy the gatekeeper hydrophobic region and made π-π stacking interaction with Phe159. The 6-position substituent at the core extended to solvent exposed region (Figure 3).

Based on the strategy of merge pharmacophore, we took the key fragment N-phenylpyrido[3,2-*d*]pyrimidin-4-amine as the recognition cap of HDAC and introduce zinc-binding motif (ZBG) through suitable linker in the 6-position of the core, designed a series of small molecule compounds simultaneously targeting both Mnk and HDAC (Figure 4).

## 2. Results and Discussion

### 2.1. Chemistry

The synthesis route of **A01** and **A07** is described in Scheme 1. Starting from commercially available 2-cyano-3-nitro-6-chloropyridine, the intermediate **1** was obtained by reduction of the nitro group with stannous chloride dihydrate which was accompanied by hydrolysis of the nitrile. Cyclization of **1** with triethylorthoformate yielded ketone **2** and chlorination with refluxing phosphorus oxychloride afforded key intermediate 4,6-dichloro pyrido[3,2-*d*]pyrimidine **3**.

The intermediate **4** was obtained by nucleophilic substitution of **3** with 4-fluoroaniline in the presence of isopropanol, then a palladium catalyzed Suzuki-coupling to furnish the carboxylic acid **5**. Subsequent treatment of the carboxylic acid with o-phenylenediamine was reacted in THF in the presence of HATU, DMAP and DIEPA to yield **A01**. Intermediate **5** was esterified and underwent ammonolysis with hydroxylamine aqueous solution under strong alkaline conditions to obtain **A07**.

The synthesis route of **A02** and **A08** is described in Scheme 2. The intermediate **7** was obtained by Knoevenagel condensation of 4-formylphenylboronic acid with malonate, then a palladium catalyzed Suzuki-coupling to obtained intermediate **8** was performed. Intermediate **8** is condensed with *o*-phenylenediamine to obtained the target compound **A02**. The intermediate **8** was methyl esterified and an aminolysis reaction successfully yielded the target compound **A08**.

Scheme 3 introduce the synthesis route of **A03** and **A09**. Intermediate **4** and methyl 4-hydroxymethylbenzoate were subjected to an Ullmann reaction to obtain intermediate **10**. The target compound **A09** was obtained by aminolysis of intermediate **10** with aqueous hydroxylamine solution. Intermediate **10** afforded the target compound **A03** by hydrolysis and condensation with *o*-phenylenediamine.

The synthesis route of target compounds **A04**–**A06** and **A10**–**A19** is shown in Scheme 4. Using 2-cyano-3-nitro-6-chloropyridine as starting material, nucleophilic substitution reactions with different amines afforded intermediates **12a**–**12c**. The intermediates **15a**–**15c** were obtained by cyclization and chlorination of **12a**–**12c**. Intermediates **15a**–**15c** underwent a C-N coupling reaction with different anilines in the presence of palladium catalyst to obtained intermediates **16a**–**16j**. The target compounds **A10**–**A19** were obtained by aminolysis reactions of intermediates **16a**–**16j**. The intermediates **16a**–**16c** afforded the target compounds **A04**–**A06** by hydrolysis and condensation with *o*-phenylenediamine.

### 2.2. Bioactivity Assay

To investigate the HDAC and Mnk inhibitory potency of the target compounds, the inhibition rate of Mnk1 and Mnk2 at 1 μM were determined using homogeneous time-resolved fluorescence assay (HTRF) and the IC_50_ value against HDAC was determined by fluorescence analysis. Compound **D06**, **Staurosporine** and **Vorinostat** were used as positive controls. The results are shown in Table 1 and Table 2.

Based on the structure of compound **D06**, we first introduced a zinc ion binding group (ZBG) through a linker with different length and rigidity to obtain the compounds **A01**–**A12**. The results showed that when the ZBG was hydroxamic acid motif, the inhibitory against Mnks and HDACs were better than *o*-phenylenediamine (**A04**, **A05**, **A06** vs. **A10**, **A11**, **A12**). The linker’s properties also have significant impact on the enzyme inhibitory activity of these compounds. When the compound linkers were rigid, the inhibitory activities of these compounds on HDACs were poor. When the linkers were replaced with a flexible alkyl chain, the potency of these compounds on Mnks and HDACs were increased significantly (**A07**–**A10** vs. **A11**, **A12**). The length of the linker also had important effect on the potency. Although when the linker was composed of a NH and five methylene units, the compound has better inhibitory activity against HDACs (**A11**, IC_50_ = 0.18 μM), but its inhibitory activity against Mnks was poor (Mnk1: 49%, Mnk2: 61%). Compared to compound **A11**, when the length of linker was a NH and six methylene groups, despite the fact the potency against HDACs declined (**A12**, IC_50_ = 0.78 μM), the inhibitory rate against Mnk was obviously improved (Mnk1: 54%, Mnk2: 70%), but the inhibitory activity of compounds **A01**–**A12** against Mnk declined differently compared with **D06**.

In order to improve the inhibitory activity of the compounds against HDAC while maintaining the Mnk activity, we fixed the linker as a flexible alkyl chain containing a NH unite and six methylene groups and introduced different substituents in the phenyl at the 4-position of the core to investigate the inhibitory activity of the compounds against Mnks and HDACs. The results are shown in Table 2, where it can be seen that when the 2-position or 3-position was substituted, the inhibitory activity of the compounds against Mnks declined (**A13**–**A19** vs. **A12**). This may be due to the increased steric hindrance when there are too many substituents on the 4-position of the benzene ring, and this may block the 4-substituted benzene ring from occupying the hydrophobic pocket. Meanwhile, the inhibitory activity against HDAC was changed differently. When the 2-position or 3-position was substituted, most compounds showed better HDAC inhibitory activity than **A12**, except for compounds **A13** and **A19**. Especially when the substituent was an electron-withdrawing group, the inhibitory activity against HDAC increased significantly (**A16**, **A17** vs. **A12**). It was worth noting that when the substituents on the benzene ring were 2-methylcarbamoyl-4-fluoro, the compound **A16** showed the best HDAC inhibitory activity among all the synthesized compounds, whose IC_50_ value against HDAC was 0.037 μM and was better than positive control **Vorinostat**. Unfortunately, the inhibitory activity of compound **A16** on Mnk declined dramatically compared to compound **A12**.

As Mnks and HDACs are overexpressed in prostate cancer, the GI_50_ values of the target compounds against PC-3 cell line were determined by an MTT assay. The results are presented in Table 3. The results showed that most of the target compounds showed moderate or excellent anti-proliferative effect on PC-3 cells. When the zinc-binding motif was hydroxamic acid, the anti-proliferative effect was better than *o*-phenylenediamine (except **A03** and **A04**). The length of the flexible linker has a significant influence on the anti-proliferative activity: compounds containing a NH unite and 6 methylene groups were greatly better than compounds containing a NH unite and 5 methylene groups (**A11** vs. **A12**). The inhibitory activity of compounds on Mnks and HDACs was closely related to their anti-proliferative activity: when the compound only showed good inhibitory activity on Mnk or HDAC, the cell activity of the compound may be poor (**A11**, **A14**, **A16**, **A17** vs. **A12**). When the substituent was 4-fluoroaniline, the compounds showed best anti-proliferative activity among the other substituents (**A13**–**A19** vs. **A12**). The anti-proliferative activity of the most potent compound **A12** (GI_50_ = 2.91 μM) was better than **D06** and equivalent to vorinostat, which should be further studied.

### 2.3. Docking Study

In order to better understand the binding mode of these compounds to Mnks and HDACs protein, we chose **A12** to conduct a molecular docking study. The results are shown in Figure 5. As illustrated in Figure 5A,B, the docking result of **A12** with Mnk2 is basically the same as **D06**: The N1 of pyrido[3,2-*d*]pyrimidine core can form conventional hydrogen bond interactions with Met162 in the hinge region. The 4-fluoroaniline can occupy the gatekeeper hydrophobic region and form a π-π stacking interaction with Phe159. The F atom can form a halogen interaction with Lys113 and Asp226 in the hydrophobic region. What’s more, the NH and OH group of the hydroxamic acid motif can form hydrogen bond interactions with Glu92 in the solvent exposed region. This may be the reason why the compounds show better Mnk inhibitory when the hydroxamic acid is substituted. The possible binding mode of compound **A12** with HDAC1 is exhibited in Figure 5C,D. Compound **A12** binds to HDAC1 by embedding the flexible chain into the hydrophobic tunnel and anchoring the terminal hydroxamic acid group at the bottom of the pocket. Besides the coordination with the zinc ion, the hydroxamic acid group also forms hydrogen bond interactions with His131, His132 and Tyr297, respectively. In addition, the NH group of the linker can form hydrogen bond interactions with Glu92 in the hydrophobic tunnel. In the solvent exposed region, the pyridine ring of the pyrido[3,2-*d*]pyrimidine core and the 4-fluoroaniline part can form π-π stacking interactions with Phe200 and the F atom can form halogen interaction with Glu201. All in all, the predicted binding mode of compound **A12** with Mnk2 and HDAC1 can further guide us to find more potent Mnk/HDAC dual inhibitors.

## 3. Materials and Methods

### 3.1. General Information

Melting points were measured with an X-4 digital micro-melting point apparatus (Shanghai Jingke Co. Ltd., Shanghai, China ) and are uncorrected. ^1^H-NMR spectra were measured with a ARX (600 MHz) spectrometer (Bruker, Karlsruhe, Germany) and ^13^C-NMR with a Bruker AV (150 MHz) instrument. Chemical shifts are recorded in δ units using tetramethylsilane as the standard (NMR peak description: s, singlet; d, doublet; t, triplet; q, quartet; m, multiplet; br, broad peak). Organic solutions were dried over anhydrous MgSO_4_ during workup. Column chromatography was carried out on a Combiflash Rf+ preparative liquid chromatograph (TELEDYNE ISCO, Lincoln, NE, USA). Silica gel 60 (200–300 mesh) and TLC were purchased from Qingdao Haiyang Chemical Co. Ltd. (Qingdao, China). All commercial reagents and solvents were used without further purification unless otherwise noted.

### 3.2. Chemistry

#### 3.2.1. Procedure for the Synthesis of **A01** and **A07**

##### *3-Amino-6-chloropicolinamide* (**1**)

2-Cyano-3-nitro-6-chloropyridine (2.0 g, 10.9 mmol), stannous chloride dihydrate (10.0 g, 44.3 mmol) and ethanol (20 mL) were added to a 50 mL pear-shaped flask and stirred at 78 °C for 2 h. After the reaction was complete, the solution was cooled to room temperature and the ethanol was evaporated to dryness. The residue was dissolved in ethyl acetate and the mixture was adjusted to pH 8 with a 2 mol/L aqueous sodium hydroxide solution, filtered and collected the organic layer, and the aqueous layer was washed twice with ethyl acetate. The organic layer was combined and washed twice with saturated sodium chloride, dried over anhydrous sodium sulfate, filtered and concentrated to obtain intermediate **1** as a yellow solid (3.0 g, yield 80.40%).

##### *6-Chloropyrido[3,2-d]pyrimidin-4(3H)-one* (**2**)

Intermediate **1** (2.5 g, 14.6 mmol) and triethyl orthoformate (100 mL, 60 mmol) were added to a 250 mL pear-shaped flask, and reacted at 146 °C for 2 h. After reaction completion was indicated by TLC, the mixture was cooled to room temperature, filtered and the solid dried to obtain intermediate **2** as a yellow solid (2.25 g, yield 85.14%).

##### *4,6-Dichloropyrido[3,2-d]pyrimidine* (**3**)

To a 100 mL pear-shaped flask intermediate **3** (1.81 g, 10 mmol), phosphorus oxychloride(36 mL) and 10 drops of *N*,*N*-dimethylaniline were added, followed by stirring for 1 h at 106 °C. The reaction was monitored by TLC. Upon completion the reaction mixture was cooled and evaporated. The obtained solid was redissolved with dichloromethane and the pH of the solution was adjusted to 8 with aqueous sodium hydrogen carbonate. The organic layer and the aqueous layer were separated, and the aqueous layer was extracted twice with dichloromethane. The organic layers were combined and dried over sodium sulfate, concentrated and purified by column chromatography on silica (petroleum ether:ethyl acetate = 100:3) to give intermediate **3** as a white solid (1.2 g, yield 60.61%).

##### *6-Chloro-N-(4-fluorophenyl)pyrido[3,2-d]pyrimidin-4-amine* (**4**)

A mixture of intermediate **3**(1.50 g, 7.54 mmol), 4-fluoroaniline (0.92 g, 8.30 mmol) and triethylamine(0.84 g, 8.30 mmol) in isopropanol(50 mL) was stirred at 50 °C for 30 min. After the reaction was completely monitored by TLC, the mixture was filtered and washed by water, then the cake was dried to obtain intermediate **4** (2.00 g, yield 96.79%).

##### *4-(4-((4-Fluorophenyl)amino)pyrido[3,2-d]pyrimidin-6-yl)benzoic acid* (**5**)

A mixture of **4** (0.21 g, 0.76 mmol), 4-phenylboronic acid (0.19 g, 1.5 mmol), bis(triphenylphosphine)palladium(II) chloride (0.028 g, 0.05 mmol) and potassium carbonate (0.32 g, 2.28 mmol) in 1,4-dioxane and water (20 mL, *V*/*V* = 4:1) was refluxed for 2 h under a N_2_ atmosphere. After the reaction completion was indicated by TLC, the mixture was cooled to room temperature and then poured to cooled water (20 mL). The pH of the mixture was adjusted to 5 using 10% aqueous hydrochloric acid solution and then it was filtered and the solid was washed with dichloromethane and then dried to obtain intermediate **5** (0.26 g, yield 94.20%).

##### *Methyl 4-(4-((4-fluorophenyl)amino)pyrido[3,2-d]pyrimidin-6-yl)benzoate* (**6**)

Intermediate **5** (0.25 g, 0.69 mmol) was placed in a 250 mL pear-shaped flask and dissolved with 60 mL methanol. Then 10 drops of concentrated sulfuric acid were added and the mixture was stirred for 4 h under reflux. After the reaction was complete, as indicated by TLC monitoring, the mixture was cooled to room temperature and then evaporated to dryness. The residue was redissolved with dichloromethane and water and brine were used to wash the mixture. The organic layer then dried over sodium sulfate, concentrated and purified by column chromatography on silica (dichloromethane:methanol = 250:1) to gave intermediate **6** as a light yellow solid (0.22 g, 84.62%)

##### *N-(2-Aminophenyl)-4-(4-((4-fluorophenyl)amino)pyrido[3,2-d]pyrimidin-6-yl)benzamide* (**A01**)

A mixture of intermediate **5** (0.26 g, 0.72 mmol), 2-(7-aza-1*H*-benzotriazole-1-yl)-1,1,3,3- tetramethyluronium hexafluorophosphate (HATU, 0.33 g, 0.86 mmol), 4-dimethylaminopyridine (DMAP, 0.11 g, 0.36 mmol) and *N*, *N*-diisopropylethylamine (DIEA, 0.28 g, 2.16 mmol) in THF (30 mL) was stirred for 30 min at room temperature, then *o*-phenylenediamine (0.093 g, 0.86 mmol) was added and the mixture stirred for 2 h. After the reaction was complete, the mixture was evaporated to dryness and the residue was redissolved with dichloromethane and 10% aqueous hydrochloric acid solution, water, saturated sodium bicarbonate solution, water and brine were used to succesivley wash the mixture, the organic layer then dried over sodium sulfate, concentrated and purified by column chromatography on silica (dichloromethane:methanol = 250:1) to gave the target compound **A01** as a light yellow solid (0.11g, yield 33.95%) (More information please see Appendix A). Mp: 211.2–213.0 °C; ^1^H-NMR (DMSO-d_6_) δ: 10.22 (s, 1H), 9.85 (s, 1H), 8.70 (d, *J* = 8.2 Hz, 2H), 8.65 (d, *J* = 8.8 Hz, 1H), 8.63 (s, 1H), 8.31 (d, *J* = 8.8 Hz, 1H), 8.19 (d, *J* = 8.2 Hz, 2H), 8.01–8.00 (m, 2H), 7.31–7.28 (m, 2H), 7.22 (d, *J* = 7.4 Hz, 1H), 7.00 (t, *J* = 7.5 Hz, 1H), 6.81 (d, *J* = 7.2 Hz, 1H), 6.63 (t, *J* = 7.3 Hz, 1H), 4.97 (s, 2H); ^13^C-NMR (DMSO-d_6_) δ: 164.9, 160.1, 157.7, 155.5, 154.1, 144.1, 143.4, 139.8, 137.0, 135.7, 134.8, 131.2, 128.4(2C), 127.6(2C), 127.0, 126.7, 125.9, 125.0, 124.9, 123.3, 116.3, 116.2, 115.3, 115.1;LC-MS *m*/*z*: 451.17 [M + H]^+^.

##### *4-(4-((4-Fluorophenyl)amino)pyrido[3,2-d]pyrimidin-6-yl)-N-hydroxybenzamide* (**A07**)

Intermediate **6** (0.1 g, 0.27 mmol) was placed in a 250 mL pear-shaped flask and dissolved with 120 mL of methanol. The mixture was stirred for 5 min at 0 °C and then the pH was adjusted to 11 by adding 2 M NaOH aqueous solution. Then 5 mL aqueous hydroxylamine solution (50%) was added and the reaction was allowed to warm to room temperature. After the reaction was complete, as monitored by TLC, the mixture was partially evaporated and the pH of the residue was adjusted to 6 by adding 10% aqueous hydrochloric acid solution. The crude **A07** was obtained by filtration and then purified by column chromatography on silica (dichloromethane:methanol = 15:1) to afford the target compound **A07** as a light yellow solid (0.06 g, yield 59.23%). Mp: 249.0–251.8 °C; ^1^H-NMR (DMSO-d_6_) δ: 11.41 (s, 1H), 10.20 (s, 1H), 9.14 (s, 1H), 8.63 (d, *J* = 8.2 Hz, 2H), 8.62 (s, 1H), 8.59 (d, *J* = 8.8 Hz, 1H), 8.29 (d, *J* = 8.8 Hz, 1H), 8.00–7.98 (m, 2H), 7.95 (d, *J* = 8.2 Hz, 2H), 7.30–7.28 (m, 2H); ^13^C-NMR (DMSO-d_6_) δ: 163.6, 160.0, 157.6, 155.4, 154.1, 144.1, 139.6, 136.9, 134.8, 133.9, 131.1, 127.7 (2C), 127.4 (2C), 125.8, 124.9, 124.8, 115.3, 115.1; LC-MS *m*/*z*: 373.9 [M − H]^−^.

#### 3.2.2. Procedure for the Synthesis of **A02** and **A08**

##### *(E)-3-(4-Boronophenyl)acrylic acid* (**7**)

A mixture of (4-acetylphenyl)boronic acid(0.10 g, 0.67 mmol), malonate (0.21 g, 0.20 mmol) and pyridine (0.053 g, 0.67 mmol) in dry toluene (10 mL) was stirred at reflux for 2 h. After the reaction was complete as indicated by TLC, the mixture was cooled to room temperature and poured into water. The pH of the mixture was adjust to 5 by adding 10% aqueous hydrochloric acid solution and then it was filtered to give intermediate 7 as a white solid (0.16 g, yield 80.73%).

##### *(E)-3-(4-(4-((4-fluorophenyl)amino)pyrido[3,2-d]pyrimidin-6-yl)phenyl)acrylic acid* (**8**)

Intermediate **8** was prepared from intermediate **7** using the same reaction conditions described above for making intermediate **5**. Light yellow solid, yield 90.0%.

##### *Methyl (E)-3-(4-(4-((4-fluorophenyl)amino)pyrido[3,2-d]pyrimidin-6-yl)phenyl)acrylate* (**9**)

Intermediate **9** was prepared from intermediate **8** using the same reaction conditions described above for making intermediate **6**. Light yellow solid, yield 80.8%.

##### *(E)-N-(2-aminophenyl)-3-(4-(4-((4-fluorophenyl)amino)pyrido[3,2-d]pyrimidin-6-yl)phenyl)acrylamide* (**A02**)

Compound **A02** was prepared from intermediate **8** using the same reaction conditions described above for making compound **A01**. Light yellow solid, yield 54.05%. Mp: 263.8–265.5 °C; ^1^H-NMR (DMSO-d_6_) δ: 10.18 (s, 1H), 9.46 (s, 1H), 8.63 (d, *J* = 8.0 Hz, 2H), 8.62 (s, 1H), 8.58 (d, *J* = 8.8 Hz, 1H), 8.28 (d, *J* = 8.8 Hz, 1H), 8.02–8.00 (m, 2H), 7.82 (d, *J* = 8.0 Hz, 2H), 7.67 (d, *J* = 15.7 Hz, 1H), 7.37 (d, *J* = 7.8 Hz, 1H), 7.31–7.28 (m, 2H), 7.05 (d, *J* = 15.7 Hz, 1H), 6.94 (t, *J* = 7.5 Hz, 1H), 6.77 (d, *J* = 7.9 Hz, 1H), 6.60 (t, *J* = 7.5 Hz, 1H), 4.98 (s, 2H); ^13^C-NMR (DMSO-d_6_) δ: 163.4, 160.0, 157.5, 155.2, 154.3, 143.9, 141.7, 138.9, 138.1, 136.8, 136.5, 134.8, 131.1, 128.3(2C), 128.1(2C), 125.9, 125.6, 124.8, 124.7, 124.7, 123.5, 123.4, 116.3, 116.1, 115.2, 115.0; LC-MS *m*/*z*: 475.17 [M − H]^−^.

##### *(E)-3-(4-(4-((4-Fluorophenyl)amino)pyrido[3,2-d]pyrimidin-6-yl)phenyl)-N-hydroxyacrylamide* (**A08**)

Compound **A08** was prepared from intermediate **9** using the same reaction conditions described above for making compound **A07**. Light yellow solid, yield 58.00%. Mp: 178.1–180.0 °C; ^1^H-NMR (DMSO-d_6_) δ: 10.80 (s, 1H), 10.16 (s, 1H), 9.16 (s, 1H), 8.62 (s, 1H), 8.58 (d, *J* = 8.2 Hz, 2H), 8.57 (d, *J* = 8.8 Hz, 1H), 8.27 (d, *J* = 8.8 Hz, 1H), 8.01–7.99 (m, 2H), 7.75 (d, *J* = 8.2 Hz, 2H), 7.56 (d, *J* = 15.5 Hz, 1H), 7.30–7.27 (m, 2H), 6.61 (d, *J* = 15.5 Hz, 1H); ^13^C-NMR (DMSO-d_6_) δ: 162.6, 160.0, 157.5, 155.2, 154.3, 144.0, 137.9, 137.5, 136.8, 136.5, 134.8, 131.1, 128.2(2C), 128.0(2C), 125.6, 124.8, 124.7, 120.3, 115.3, 115.1; LC-MS *m*/*z*: 402.15 [M + H]^+^.

#### 3.2.3. Procedure for the Synthesis of **A03** and **A09**

##### *Methyl 4-(((4-((4-fluorophenyl)amino)pyrido[3,2-d]pyrimidin-6-yl)oxy)methyl)benzoate* (**10**)

A mixture of intermediate **4** (0.69 g, 2.52 mmol), methyl 4-(hydroxymethyl)benzoate (0.4 g, 0.21 mmol), cuprous iodide (0.048 g, 0.25 mmol), 8-hydroxyquinoline (0.073 g, 0.50 mmol) and cesium carbonate (0.16 g, 5 mmol) in DMF (20 mL) was stirred at 110 °C under a N_2_ atmosphere. After the reaction was complete, as indicated by TLC monitoring, the mixture was cooled to room temperature and the DMF was evaporated to dryness. The residue was redissolved with dichloromethane and water and brine were used to wash the mixture. The organic layer was then dried over sodium sulfate, concentrated and purified by column chromatography on silicagel (petroluem ether:ethyl acetate = 5:1) to gave the intermediate **10** as a yellow solid (0.54 g, yield 53.04%).

##### *4-(((4-((4-Fluorophenyl)amino)pyrido[3,2-d]pyrimidin-6-yl)oxy)methyl)benzoic acid* (**11**)

Intermediate **10** was placed in a 50 mL pear-shaped flask and dissolved with 10 mL of DMF. Then 10 mL 10% aqueous sodium hydroxide solution was added to the mixture that was then stirred at 60 °C for 4 h. After the reaction was complete as monitored by TLC, the mixture was poured to 100 mL of water and the pH of the mixture was adjusted to 5 using 10% aqueous hydrochloric acid solution and then filtered to gave the intermediate **11** as a light yellow solid (0.092 g, yield 95.3%).

##### *N-(2-Aminophenyl)-4-(((4-((4-fluorophenyl)amino)pyrido[3,2-d]pyrimidin-6-yl)oxy)methyl)benzamide* (**A03**)

Compound **A03** was prepared from intermediate **11** using the same reaction conditions described above for making compound **A01**. Light yellow solid, yield 75.0%. Mp: 216.3–218.0 °C; ^1^H-NMR (DMSO-d_6_) δ: 9.66 (s, 1H), 9.61 (s, 1H), 8.56 (s, 1H), 8.15 (d, *J* = 9.0 Hz, 1H), 8.02 (d, *J* = 7.8 Hz, 2H), 8.00–7.97 (m, 2H), 7.71 (d, *J* = 7.8 Hz, 2H), 7.47 (d, *J* = 9.0 Hz, 1H), 7.30–7.27 (m, 2H), 7.16 (d, *J* = 7.6 Hz, 1H), 6.97 (t, *J* = 7.5 Hz, 1H), 6.78 (d, *J* = 7.9 Hz, 1H), 6.59 (t, *J* = 7.4 Hz, 1H), 5.82 (s, 2H), 4.90 (s, 2H); ^13^C-NMR (DMSO-d_6_) δ: 165.1, 160.7, 159.7, 157.4, 156.1, 153.2, 143.2, 142.0, 140.4, 139.6, 135.0, 134.3, 128.0(2C), 128.0(2C), 126.8, 126.6, 124.0, 123.9, 123.3, 119.8, 116.2(2C), 115.3, 115.1, 67.7; LC-MS *m*/*z*: 481.20 [M + H]^+^.

##### *4-(((4-((4-Fluorophenyl)amino)pyrido[3,2-d]pyrimidin-6-yl)oxy)methyl)-N-hydroxybenzamide* (**A09**)

Compound **A09** was prepared from intermediate **10** using the same reaction conditions described above for making compound **A07**. Light yellow solid, yield 63.5%. Mp: 169.8–171.1 °C; ^1^H-NMR (DMSO-d_6_) δ: 11.22 (s, 1H), 9.58 (s, 1H), 9.03 (s, 1H), 8.55 (s, 1H), 8.14 (d, *J* = 9.0 Hz, 1H), 7.97–7.95 (m, 2H), 7.79 (d, *J* = 8.0 Hz, 2H), 7.64 (d, *J* = 8.0 Hz, 2H), 7.45 (d, *J* = 9.0 Hz, 1H), 7.29–7.26 (m, 2H), 5.77 (s, 2H); ^13^C-NMR (DMSO-d_6_) δ: 164.0, 160.7, 159.7, 157.4, 156.1, 153.2, 142.0, 140.2, 139.6, 135.0, 132.4, 128.1 (2C), 127.9, 127.1, 124.0, 123.9, 119.8, 115.3, 115.1, 67.7; LC-MS *m*/*z*: 406.1 [M + H]^+^.

#### 3.2.4. General Procedure for the Synthesis of **A04**–**A06**

##### General Procedure for the Synthesis of **12a**–**12c**

A mixture of 2-cyano-3-nitro-6-chloropyridine (1.0 equivalent), the required amine (1.5 equivalents) and *N*,*N*-dimethylaniline (3 equivalents) in 10 mL of DMF was stirred at 60 °C for 2 h. After the reaction was complete as monitored by TLC, the mixture was cooled to room temperature, then poured into 100 mL water, stirred for 10 min and filtered to gave intermediates **12a**–**12c**.

##### General Procedure for the Synthesis of **13a**–**13c**

Intermediates **13a**–**13c** were prepared from **12a**–**12c** using the same reaction conditions described above for making intermediate **1**.

##### General Procedure for the Synthesis of **14a**–**14c**

Intermediates **14a**–**14c** were prepared from **13a**–**13c** using the same reaction conditions described above for making intermediate **2**.

##### General Procedure for the Synthesis of **15a**–**15c**

Intermediates **15a**–**15c** were prepared from **14a**–**14c** using the same reaction conditions described above for making intermediate **3**.

##### General Procedure for the Synthesis of **16a**–**16j**

A mixture of intermediate **15a** (0.61 mmol), the required aniline (0.73 mmol), tris(dibenzylideneacetone)dipalladium (Pd_2_(dba)_3_, 0.03 mmol), 2-dicyclohexylphosphino-2′,6′-di-isopropoxy-1,1′-bipheny(RuPhos, 0.06 mmol) and potassium carbonate (0.06 mmol) in 20 mL of 1,4-dioxane was stirred at 90 °C under a N_2_ atmosphere. After the reaction was complete, as monitored by TLC, the mixture was cooled to room temperature and the 1,4-dioxane was evaporated to dryness. The residue was redissolved with dichloromethane and washed with water and brine. The organic layer then dried over sodium sulfate, concentrated and purified by column chromatography on silica (dichloromethane:methanol = 125:1) to gave intermediates **16a**–**16j** as light yellow solids (yield 48.00–73.24%).

##### General Procedure for the Synthesis of **17a**–**17c**

Intermediates **17a**–**17c** were prepared from **16a**–**16c** using the same reaction conditions described above for making intermediate **11**.

##### *N-(2-Aminophenyl)-4-(((4-((4-fluorophenyl)amino)pyrido[3,2-d]pyrimidin-6-yl)amino)methyl)benzamide* (**A04**)

Compound **A04** was prepared from intermediate **17a** using the same reaction conditions described above for making compound **A01**. Light yellow solid, yield 69.10%. Mp: 202.8–204.1 °C; ^1^H-NMR (DMSO-d_6_) δ: 9.60 (s, 1H), 9.17 (s, 1H), 8.37 (s, 1H), 8.10 (t, *J* = 5.4 Hz, 1H), 7.98–7.95 (m, 4H), 7.80 (d, *J* = 9.1 Hz, 1H), 7.61 (d, *J* = 7.9 Hz, 2H), 7.25–7.22 (m, 2H), 7.19 (d, *J* = 9.1 Hz, 1H), 7.14 (d, *J* = 7.6 Hz, 1H), 6.95 (t, *J* = 7.5 Hz, 1H), 6.77 (d, *J* = 7.9 Hz, 1H), 6.58 (t, *J* = 7.4 Hz, 1H), 4.88 (d, *J* = 5.6 Hz, 2H), 4.86 (s, 2H); ^13^C-NMR (DMSO-d_6_) δ: 165.2, 159.2, 156.8, 156.3, 154.9, 150.3, 143.9, 143.2, 139.4, 136.5, 135.5, 133.1, 129.4, 127.9(2C), 127.5(2C), 126.7, 126.5, 123.4, 122.5, 119.3, 116.2(2C), 115.3, 115.1, 44.1; LC-MS *m*/*z*: 480.21 [M + H]^+^.

##### *N-(2-Aminophenyl)-6-((4-((4-fluorophenyl)amino)pyrido[3,2-d]pyrimidin-6-yl)amino)hexanamide* (**A05**)

Compound **A05** was prepared from intermediate **17b** using the same reaction conditions described above for making compound **A01**. Pale brown solid, yield 67.40%. Mp: 209.5–211.5 °C; ^1^H-NMR (DMSO-d_6_) δ: 9.16 (s, 1H), 9.08 (s, 1H), 8.35 (s, 1H), 7.97–7.95 (m, 2H), 7.74 (d, *J* = 9.1 Hz, 1H), 7.44 (s, 1H), 7.21–7.18 (m, 2H), 7.14 (d, *J* = 7.8 Hz, 1H), 7.09 (d, *J* = 9.1 Hz, 1H), 6.88 (t, *J* = 8.3 Hz, 1H), 6.70 (d, *J* = 8.0 Hz, 1H), 6.51 (t, *J* = 8.1 Hz, 1H), 4.82 (s, 2H), 3.56 (q, *J* = 6.6 Hz, 2H), 2.35 (t, *J* = 7.4 Hz, 2H), 1.73–1.63 (m, 4H), 1.51–1.45 (m, 2H); ^13^C-NMR (DMSO-d_6_) δ: 171.2, 159.1, 156.7, 154.9, 149.9, 141.9, 139.1, 136.1, 135.6, 129.6, 125.8, 125.4, 123.6, 122.6(2C), 119.4, 116.2, 116.0, 115.3, 115.1, 40.5, 35.9, 28.7, 26.5, 25.3; LC-MS *m*/*z*: 460.24 [M + H]^+^.

##### *N-(2-Aminophenyl)-7-((4-((4-fluorophenyl)amino)pyrido[3,2-d]pyrimidin-6-yl)amino)heptanamide* (**A06**)

Compound **A06** was prepared from intermediate **17c** using the same reaction conditions described above for making compound **A01**. Pale brown solid, yield 65.40%. Mp: 125.6–127.0 °C; ^1^H-NMR (DMSO-d_6_) δ: 9.15 (s, 1H), 9.07 (s, 1H), 8.35 (s, 1H), 7.97–7.95 (m, 2H), 7.74 (d, *J* = 9.1 Hz, 1H), 7.42 (s, 1H), 7.23–7.20 (m, 2H), 7.14 (d, *J* = 6.7 Hz, 1H), 7.09 (d, *J* = 9.1 Hz, 1H), 6.88 (t, *J* = 8.3 Hz, 1H), 6.70 (d, *J* = 7.9 Hz, 1H), 6.52 (t, *J* = 7.5 Hz, 1H), 4.81 (s, 2H), 3.54 (q, *J* = 6.6 Hz, 2H), 2.32 (t, *J* = 7.4 Hz, 2H), 1.64–1.61 (m, 4H), 1.50–1.44 (m, 2H), 1.43–1.39 (m, 2H); ^13^C-NMR (DMSO-d_6_) δ: 171.2, 159.2, 156.7, 154.9, 149.9, 141.9, 139.1, 136.1, 135.5, 129.5, 125.7, 125.3, 123.7, 122.5(2C), 119.4, 116.2, 116.0, 115.3, 115.1, 40.6, 35.8, 28.7(2C), 26.6, 25.4; LC-MS *m*/*z*: 474.28 [M + H]^+^.

#### 3.2.5. General Procedure for the Synthesis of **A10**–**A19**

##### *4-(((4-((4-Fluorophenyl)amino)pyrido[3,2-d]pyrimidin-6-yl)amino)methyl)-N-hydroxybenzamide* (**A10**)

Compound **A10** was prepared from intermediate **16a** using the same reaction conditions described above for making compound **A07**. Mp: 197.7–199.3 °C; ^1^H-NMR (DMSO-d_6_) δ: 11.15 (s, 1H), 9.14 (s, 1H), 8.96 (s, 1H), 8.36 (s, 1H), 8.08 (t, *J* = 5.5 Hz, 1H), 7.96–7.94 (m, 2H), 7.79 (d, *J* = 9.1 Hz, 1H), 7.73 (d, *J* = 8.2 Hz, 2H), 7.54 (d, *J* = 8.2 Hz, 2H), 7.24–7.21 (m, 2H), 7.18 (d, *J* = 9.1 Hz, 1H), 4.83 (d, *J* = 5.7 Hz, 2H); ^13^C-NMR (DMSO-d_6_) δ: 164.2, 158.9, 157.3, 156.3, 155.0, 150.1, 143.7, 139.0, 136.2, 135.4, 131.3, 129.3, 127.6(2C), 127.0(2C), 122.7(2C), 115.3, 115.2, 44.05; LC-MS *m*/*z*: 405.2 [M + H]^+^.

##### *6-((4-((4-Fluorophenyl)amino)pyrido[3,2-d]pyrimidin-6-yl)amino)-N-hydroxyhexanamide* (**A11**)

Compound **A11** was prepared from intermediate **16b** using the same reaction conditions described above for making compound **A07**. Mp: 187.6–189.5 °C; ^1^H-NMR (DMSO-d_6_) δ: 10.34 (s, 1H), 9.15 (s, 1H), 8.66 (s, 1H), 8.35 (s, 1H), 7.98–7.96 (m, 2H), 7.74 (d, *J* = 9.1 Hz, 1H), 7.42 (s, 1H), 7.24–7.21 (m, 2H), 7.08 (d, *J* = 9.1 Hz, 1H), 3.52 (q, *J* = 5.5 Hz, 2H), 1.97 (t, *J* = 7.3 Hz, 2H), 1.65–1.50 (m, 4H), 1.43–1.35 (m, 2H); ^13^C-NMR (DMSO-d_6_) δ: 169.0, 159.2, 156.7, 154.9, 149.9, 139.1, 136.1, 135.6, 129.6, 122.5(2C), 119.4, 115.3, 115.1, 40.5, 32.3, 28.5, 26.4, 25.1; LC-MS *m*/*z*: 385.20 [M + H]^+^.

##### *7-((4-((4-Fluorophenyl)amino)pyrido[3,2-d]pyrimidin-6-yl)amino)-N-hydroxyheptanamide* (**A12**)

Compound **A12** was prepared from intermediate **16c** using the same reaction conditions described above for making compound **A07**. Mp: 182.6–184.2 °C; ^1^H-NMR (DMSO-d_6_) δ: 10.33 (s, 1H), 9.14 (s, 1H), 8.66 (s, 1H), 8.35 (s, 1H), 7.98–7.95 (m, 2H), 7.74 (d, *J* = 9.1 Hz, 1H), 7.41 (s, 1H), 7.24–7.21 (m, 2H), 7.09 (d, *J* = 9.1 Hz, 1H), 3.52 (q, *J* = 6.3 Hz, 2H), 1.95 (t, *J* = 7.4 Hz, 2H), 1.63–1.58 (m, 2H), 1.54–1.49 (m, 2H), 1.43–1.38 (m, 2H), 1.35–1.28 (m, 2H); ^13^C-NMR (DMSO-d_6_) δ: 169.1, 159.2, 156.7, 156.2, 154.9, 149.9, 139.1, 136.1, 135.6, 129.6, 122.5, 119.4, 115.3, 115.1, 107.2, 40.6, 32.3, 28.6, 26.5, 25.2; LC-MS *m*/*z*: 397.0 [M − H]^−^.

##### *7-((4-((4-Fluoro-2-isopropoxyphenyl)amino)pyrido[3,2-d]pyrimidin-6-yl)amino)-N-hydroxyheptanamide* (**A13**)

Compound **A13** was prepared from intermediate **16d** using the same reaction conditions described above for making compound **A07**. Mp: 191.3–193.0 °C; ^1^H-NMR (DMSO-d_6_) δ: 10.31 (s, 1H), 9.42 (s, 1H), 8.93–8.86 (m, 1H), 8.63 (s, 1H), 8.42 (s, 1H), 7.76 (d, *J* = 8.9 Hz, 1H), 7.53 (s, 1H), 7.11 (d, *J* = 8.4 Hz, 1H), 7.11–7.09 (m, 1H), 6.84 (t, *J* = 7.5 Hz, 1H), 4.88–4.79 (m, 1H), 3.47 (q, *J* = 4.9 Hz, 2H), 1.94 (t, *J* = 6.8 Hz, 2H), 1.65–1.63 (m, 2H), 1.53–1.48 (m, 2H), 1.40–1.37 (m, 8H), 1.30–1.29 (m, 2H); ^13^C-NMR (DMSO-d_6_) δ: 169.1, 158.8, 156.6, 154.1, 150.1, 147.1, 138.9, 136.3, 129.8, 125.9, 119.6, 119.0, 106.3, 101.5, 71.5, 40.7, 32.3, 28.6, 28.5, 26.6, 25.2, 21.9(2C); LC-MS *m*/*z*: 457.24 [M + H]^+^.

##### *7-((4-((4-Fluoro-2-methoxyphenyl)amino)pyrido[3,2-d]pyrimidin-6-yl)amino)-N-hydroxyheptanamide* (**A14**)

Compound **A14** was prepared from intermediate **16e** using the same reaction conditions described above for making compound **A07**. Mp: 205.9–207.6 °C; ^1^H-NMR (DMSO-d_6_) δ: 10.32 (s, 1H), 9.45 (s, 1H), 8.74 (dd, *J* = 8.9, 6.4 Hz, 1H), 8.63 (s, 1H), 8.43 (s, 1H), 7.76 (d, *J* = 9.1 Hz, 1H), 7.63 (s, 1H), 7.10 (d, *J* = 9.1 Hz, 1H), 7.08 (dd, *J* = 10.6, 2.7 Hz, 1H), 6.86 (td, *J* = 8.7, 2.7 Hz, 1H), 3.96 (s, 3H), 3.43 (q, *J* = 6.7 Hz, 2H), 1.94 (t, *J* = 7.4 Hz, 2H), 1.68–1.64 (m, 2H), 1.54–1.50 (m, 2H), 1.44–1.39 (m, 2H), 1.35–1.30 (m, 2H); ^13^C-NMR (DMSO-d_6_) δ: 169.2, 159.1, 156.7, 154.3, 149.9, 149.5, 138.2, 135.9, 129.7, 124.8, 119.6, 119.3, 106.3, 99.7, 56.7, 40.8, 32.3, 28.5, 28.3, 26.6, 25.2; LC-MS *m*/*z*: 429.21 [M + H]^+^.

##### *7-((4-((2-Chloro-4-fluorophenyl)amino)pyrido[3,2-d]pyrimidin-6-yl)amino)-N-hydroxyheptanamide* (**A15**)

Compound **A15** was prepared from intermediate **16f** using the same reaction conditions described above for making compound **A07**. Mp: 203.9–205.1 °C; ^1^H-NMR (DMSO-d_6_) δ: 10.32 (s, 1H), 9.50 (s, 1H), 8.85 (dd, *J* = 9.1, 5.8 Hz, 1H), 8.63 (s, 1H), 8.45 (s, 1H), 7.79 (d, *J* = 9.1 Hz, 1H), 7.69 (s, 1H), 7.60 (dd, *J* = 8.4, 2.7 Hz, 1H), 7.34 (td, *J* = 8.8, 2.8 Hz, 1H), 7.12 (d, *J* = 9.1 Hz, 1H), 3.43 (q, *J* = 6.5 Hz, 2H), 1.94 (t, *J* = 7.4 Hz, 2H), 1.67–1.61 (m, 2H), 1.53–1.48 (m, 2H), 1.40–1.35 (m, 2H), 1.32–1.27 (m, 2H); ^13^C-NMR (DMSO-d_6_) δ: 169.1, 156.7, 154.3, 149.7, 139.0, 136.2, 132.5, 129.6, 123.2, 121.9, 120.1, 116.8, 116.5, 115.0, 40.8, 32.3, 28.6, 28.4, 26.6, 25.3; LC-MS *m*/*z*: 433.14 [M + H]^+^.

##### *5-Fluoro-2-((6-((7-(hydroxyamino)-7-oxoheptyl)amino)pyrido[3,2-d]pyrimidin-4-yl)amino)-N-methylbenzamide* (**A16**)

Compound **A16** was prepared from intermediate **16 g** using the same reaction conditions described above for making compound **A07**. Mp: 185.5–187.0 °C; ^1^H-NMR (DMSO-d_6_) δ: 12.14 (s, 1H), 10.30 (s, 1H), 9.17 (dd, *J* = 9.3, 5.4 Hz, 1H), 8.70 (d, *J* = 4.3 Hz, 1H), 8.62 (s, 1H), 8.42 (s, 1H), 7.74 (d, *J* = 9.1 Hz, 1H), 7.57 (dd, *J* = 9.5, 3.0 Hz, 1H), 7.48-7.45 (m, 1H), 7.45-7.42 (m, 1H), 7.07 (d, *J* = 9.1 Hz, 1H), 3.64 (q, *J* = 6.8 Hz, 2H), 2.83 (d, *J* = 4.5 Hz, 3H), 1.93 (t, *J* = 7.4 Hz, 2H), 1.66–1.61 (m, 2H), 1.52–1.47 (m, 2H), 1.48–1.42 (m, 2H), 1.35–1.20 (m, 2H); ^13^C-NMR (DMSO-d_6_) δ: 169.2, 167.6, 156.6, 154.7, 149.7, 139.3, 136.1, 130.2, 122.8, 121.9, 119.4, 118.3, 118.1, 114.8, 114.5, 40.5, 32.4, 29.0, 28.7, 26.6, 26.1, 25.3; LC-MS *m*/*z*: 456.22 [M + H]^+^.

##### *7-((4-((2,4-Difluorophenyl)amino)pyrido[3,2-d]pyrimidin-6-yl)amino)-N-hydroxyheptanamide* (**A17**)

Compound **A17** was prepared from intermediate **16h** using the same reaction conditions described above for making compound **A07**. Mp: 188.8-191.7 °C; ^1^H-NMR (DMSO-d_6_) δ: 10.32 (s, 1H), 9.13 (s, 1H), 8.63 (s, 1H), 8.42–8.35 (m, 2H), 7.77 (d, *J* = 9.1 Hz, 1H), 7.57 (s, 1H), 7.44–7.39 (m, 1H), 7.18–7.15 (m, 1H), 7.11 (d, *J* = 9.1 Hz, 1H), 3.44 (q, *J* = 6.6 Hz, 2H), 1.95 (t, *J* = 7.4 Hz, 2H), 1.64–1.59 (m, 2H), 1.54–1.49 (m, 2H), 1.42–1.37 (m, 2H), 1.33–1.28 (m, 2H); ^13^C-NMR (DMSO-d_6_) δ: 169.2, 156.8, 155.1, 149.8, 138.7, 135.8, 129.5, 124.5, 123.8, 119.9, 111.4, 111.2, 104.3, 104.1, 40.7, 32.3, 28.6, 28.5, 26.6, 25.2; LC-MS *m*/*z*: 417.20 [M + H]^+^.

##### *7-((4-((4-Fluoro-3-methoxyphenyl)amino)pyrido[3,2-d]pyrimidin-6-yl)amino)-N-hydroxyheptanamide* (**A18**)

Compound **A18** was prepared from intermediate **16i** using the same reaction conditions described above for making compound **A07**. Mp: 178.6–180.0 °C; ^1^H-NMR (DMSO-d_6_) δ: 10.32 (s, 1H), 9.10 (s, 1H), 8.64 (s, 1H), 8.37 (s, 1H), 7.83 (dd, *J* = 7.9, 2.2 Hz, 1H), 7.74 (d, *J* = 9.0 Hz, 1H), 7.56–7.52 (m, 1H), 7.41 (s, 1H), 7.23–7.19 (m, 1H), 7.10 (d, *J* = 9.1 Hz, 1H), 3.88 (s, 3H), 3.53 (q, *J* = 6.1 Hz, 2H), 1.95 (t, *J* = 7.3 Hz, 2H), 1.63–1.58 (m, 2H), 1.54–1.49 (m, 2H), 1.44–1.39 (m, 2H), 1.35–1.29 (m, 2H); ^13^C-NMR (DMSO-d_6_) δ: 169.2, 156.7, 154.9, 149.9, 148.3, 146.9, 146.7, 139.2, 136.2, 136.0, 129.6, 115.6, 112.7, 107.0, 56.1, 40.6, 32.3, 28.7, 28.6, 26.5, 25.2; LC-MS *m*/*z*: 429.23 [M + H]^+^.

##### *7-((4-((3-Chloro-4-fluorophenyl)amino)pyrido[3,2-d]pyrimidin-6-yl)amino)-N-hydroxyheptanamide* (**A19**)

Compound **A19** was prepared from intermediate **16j** using the same reaction conditions described above for making compound **A07**. Mp: 207.9–208.5 °C; ^1^H-NMR (DMSO-d_6_) δ: 10.39 (s, 1H), 9.25 (s, 1H), 8.56 (s, 1H), 8.40 (s, 1H), 8.36 (dd, *J* = 6.8, 2.6 Hz, 1H), 7.95–7.92 (m, 1H), 7.74 (d, *J* = 9.1 Hz, 1H), 7.53 (s, 1H), 7.43 (t, *J* = 9.1 Hz, 1H), 7.14 (d, *J* = 9.1 Hz, 1H), 3.54 (q, *J* = 6.5 Hz, 2H), 1.96 (t, *J* = 7.4 Hz, 2H), 1.62–1.57 (m, 2H), 1.54–1.49 (m, 2H), 1.44–1.39 (m, 2H), 1.34–1.29 (m, 2H); ^13^C-NMR (DMSO-d_6_) δ: 169.2, 156.8, 154.7, 153.7, 152.1, 149.7, 139.4, 136.6, 136.1, 129.6, 122.0, 121.1, 119.0, 116.8, 116.6, 40.6, 32.3, 28.7, 26.5, 25.2; LC-MS *m*/*z*: 433.16 [M + H]^+^.

### 3.3. Biological Assays

#### 3.3.1. Anti-Proliferative Assay

The human prostatic cancer PC-3 cell line was cultured in RPMI1640 with 10% (*v*/*v*) heat-inactivated fetal bovine serum. The anti-proliferative activity of synthetic compounds were evaluated by the methylthiazolyltetrazolium bromide (MTT) assay. Cells (2 × 10^4^ cells/well) were incubated in a humidified atmosphere with 5% CO_2_ at 37 °C for 24 h, then various concentrations of each compound mixed in 100 mL medium were added to each well. After incubated for another 96 h, MTT solution (50 μL of 2 mg/mL) was added to each well and incubated for an additional 4 h. The medium was removed by aspiration and the cells were dissolved in 200 mL DMSO. The absorbance at 570 nm was measured in the 96-well plate reader. Growth inhibition was calculated and expressed as the ratio of the cell number in the treated group to that of the untreated group. The GI_50_ values were calculated according to inhibition ratios. All experiments were performed three times independently, and the results were reported presented as the mean.

#### 3.3.2. HDACs Inhibition Fluorescence Assay

The general procedure for the enzyme assay was carried out according to the published protocol [19] with slight modification. For the HDACs assay, HeLa nuclear extracts (Enzo Life Sciences, NY, USA) were used as a source of histone deacetylase. All reactions were performed in the black half area 96-well microplates. A serial dilution of the inhibitors and enzymes were pre-incubated at 25 °C for 15 min, and then fluorogenic substrate Boc-Lys(Ac)-AMC was added. After incubation at 37 °C for 60 min, the mixture was stopped by the addition of 25 mL of developer containing trypsin and vorinostat for 10 min. Fluorescence intensity was measured using a Varioskan Flash Station (Thermo Scientific, MA, USA) at excitation and emission wavelengths of 355 (or 360 for the HeLa NuEx) and 460 nm, respectively. The IC_50_ values were extracted by curve fitting the dose/response slopes.

#### 3.3.3. Mnks Inhibition HTRF Assay

Staurosporine was used as positive control in the biological evaluation assay and was obtained from Selleck (Shanghai, China). Ser/Thr KinEase assay kit (CisBio, Shanghai, China) was used to determine the Mnk1/2 kinase inhibition of the synthesized compounds using staurosporine as positive control. 1.00 ng of Mnk1 or Mnk2 was incubated with different concentrations (10,000 nM, 1000 nM, 100 nM, 10 nM, and 1 nM) of test compounds in an 10 µL reaction mixture (Mnk1: 1 µM substrate S1, 10 mM MgCl_2_, 121 µM ATP, 1 × KinEASE enzymatic buffer and 1 mM dithiothreitol (DTT Mnk2: 1 µM substrate S1, 10 mM MgCl_2_, 39.2 µM ATP, 1 × KinEASE enzymatic buffer and 1 mM DTT for 60 min at 25 °C. The mixture of 5 µL SA-XL665 and 5 µL STK Ab detection reagents following the kit protocol was added to terminate the reaction. The ratio between the HTRF signals of 615 and 665 nm recorded with an Infinite^®^ F500 microplate reader (Tecan, Männedorf, Switzerland).

### 3.4. Docking Study

Discovery Studio 3.0 (BIOVIA, Waltham, MA, USA) was used to perform in silico docking. The Xray crystal structure of Mnk and HDAC1 in complex with the ligand was retrieved from Protein Data Bank (PDB code 2HW7 and 1C3S). All calculations and manipulations were performed with CDOCKER modules in the Discovery Studio software package. All water molecules were removed and hydrogen was added. By applying the default parameters, the best docking result was selected to analysis based on the favorable binding affinity rank in kcal/mol.

## 4. Conclusions

We report herein a series of 4,6-disubstituted pyrido[3,2-*d*]pyrimidine derivatives as Mnk/HDAC dual inhibitors. The structure-activity relationships of these compounds was summarised. The vitro enzymatic assay result showed that most of the synthesized compounds showed good potency against Mnks and HDACs. Among the compounds, compound **A12** showed the best inhibitory against Mnks and also good potency on HDAC, whose Mnk1 and Mnk2 inhibitory activity with the inhibition rate of 54% and 70% at the concentration of 1 μM and the HDAC inhibitory activity with the IC_50_ value of 0.78 μM. Most of the compounds showed preferable anti-proliferation activity against human prostate cancer PC-3 cell lines with the GI_50_ value of 2.91 μM–18.26 μM. The most potent compound **A12** was selected to conduct molecular docking studies. Docking results showed that the hydroxamic acid motif could form hydrogen bond interactions with both Mnk and HDAC. The pyrido[3,2-*d*]pyrimidine core and the 4-substituted aniline group were also essential for forming key interactions between the compounds and targets. A series of novel Mnk/HDAC inhibitors bearing pyrido[3,2-*d*]pyrimidine scaffold were provided and the further research was still needed.

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
