# Peer review of "Design, Synthesis and Bioactivity Evaluation of 4,6-Disubstituted Pyrido[3,2-d]pyrimidine Derivatives as Mnk and HDAC Inhibitors"

_molecules, 2020, doi:10.3390/molecules25184318_

Round 1
Reviewer 1 Report
Minor revision:
- Insert in the introduction the concept of the multi/dual target. (https://doi.org/10.1016/j.ejmech.2019.111579, https://doi.org/10.1016/j.ejmech.2019.111903 ,https://doi.org/10.3389/fchem.2018.00130, https://doi.org/10.3390/antiox9090775)
- Abbreviate line 78-88 or include this part in discussions
- Figure 3: enlarge the label because they are small and illegible
- Calculate the binding affinity (in kcal/mol) of the compounds after the docking.
- Figure 5: enlarge the label because they are small and illegible. Are non-polar hydrogens necessary?
Suggestions:
- Insert a space after the brackets: for example line 73 “VEGFR (Figure 2)". Check all work.
- In vitro and in silico in italics
- Line 522: 2x104 cell/well is correct?
- Line 535: 37 C is 37°C?
- Insert a space between paragraphs
- Line 207: after D06 miss the point
- Line 210 and 2020: Is correct π-π stack? It would be better to write stacking interaction or π-π
Author Response
Please see the attachment.
Best regard!
Mr Kun, Xing

Reviewer 2 Report
Review of Design, Synthesis and Bioactivity Evaluation of 4,6- 2 disubstituted pyrido[3,2-d]pyrimidine Derivatives as 3 Mnk and HDAC inhibitors by Xing et al.
The MNKs have emerged as potential therapeutic targets in oncology and there are now substantial efforts to identify small molecule inhibitors of these enzymes. A number of HDAC inhibitors show a potency as promising antitumor agents with several drug candidates currently in phase I–III clinical trials. The research of potential inhibitors of these molecules is also of great interest.
Based on their previous results the authors proposed a set of derivative molecules and a study of their bioactivities. Although data presented here are clear, the logic of the ms is questionable as molecular docking of the D06 molecule probably gave clues to modify efficiently this molecule and should be introduced first.
While this represents a significant achievement, there are multiple important issues with the work prior acceptance in Molecules, as detailed below.
Major comments:
The overall ms would benefit from close English language proof-reading.
The authors need to check the 4 schemes and the text for the correspondence of compounds names, and also the substituents of compound A17 (page 5, line 178).
The inhibitory activities of the molecules towards Mnk are never compared to the Staurosporine control for the results presented in Tables 1 and 2. Moreover, the authors report the IC50 for this control, why not the % inhibition at 1µM?
A number of references are missing. As an exemple the HDAC inhibition fluorescence assay is used by many other groups.
Even if there is an overall conclusion, a discussion of the obtained results is missing.
Author Response

(The authors gave the same response as above.)

Reviewer 3 Report
The work performed by D. Liu, L. Zhao et al is about the design, synthesis and biological evaluation of a series of compounds displaying a dual inhibition mode against anticancer targets Mnk and HDAC. Compounds display a
pyrido[3,2-d]pyrimidine framework. The work is interesting and useful for the scientific community. Some errors should be corrected before publication.
Minor errors:
- In abstract says “thus simultaneously inhibit HDAC and Mnk can increase the inhibition” but should say “thus simultaneous inhibition of HDAC and Mnk can increase the inhibition”.
- In abstract says “synthesized a series 4,6-disubstituted pyrido[3,2-d]pyrimidine derivatives” but should say “synthesized a series of 4,6-disubstituted pyrido[3,2-d]pyrimidine derivatives”.
- In abstract and page#9 rows#201-202 says “was a potent Mnk /HDAC inhibitor and would futher research.” but should say “was a potent Mnk /HDAC inhibitor and will be further researched.”
- In page#1 row#24 says “plays important role” but should say “plays an important role”.
- In page#1 row#32 says “has a unsatisfactory effect” but should say “has an unsatisfactory effect”.
- In page#2 row#35 says “and other target show significant therapeutic effect than” but should say “and another target would show a higher therapeutic effect than”.
- A space between numeric values and unit symbols should be indicated, for example in page#2 rows#40-41 says “4.4nM, 2.4nM and 15.7nM” but should say “4.4 nM, 2.4n M and 15.7 nM”.
- In page#2 row#50 says “kinases(Mnk1 and Mnk2)” but should say “kinases (Mnk1 and Mnk2)”.
- In page#3 row#73 says “VEGF(Figure 2).” but should say “VEGF (Figure 2).”
- Chemical structure of cpd D06 should be drawn in Figure 2 or in Figure 3.
- In page#3 rows#80-83 says “Molecular docking result showed that the pyrido[3,2-d]pyrimidine core could imitate the structure of purine in ATP and made hydrogen bond interaction with Met162 in hinge.” but should say “Molecular docking performed between D06 and Mnk2 (PDB code 2hw7) showed that the pyrido[3,2-d]pyrimidine core could imitate the structure of purine in ATP and made hydrogen bond interaction with Met162 in hinge.”
- In page#3 row#84 says “region(Figure 3).” but should say “region (Figure 3).”
- In page#4 row#87 says “HDAC(Figure 4).” but should say “HDAC (Figure 4).”
- In Figure 3 legend says “with Mnk2(2hw7).” but should say “with Mnk2.”.
- In page#5 row#104 says “DIEPA to yield A01. Intermediate 6 was methyl esterification and underwent ammonolysis” but should say “DIPEA to yield A01. Intermediate 6 was esterified and underwent ammonolysis”.
- In Scheme 1, 2 and 4 legends should say “DIPEA” but not “DIEA”.
- In page#5 row#112 says “in Scheme 2.The intermediate 7” but should say “in Scheme 2. The intermediate 7”.
- For the preparation of 7, authors say malonate in the text and Scheme 2 legend but in the experimental part says succinic acid. It should be corrected to the right chemical name in all parts.
- In the Experimental part, a space should be always indicated between the name of the reagent and the parentheses which indicate amounts.
- In page#6 rows#124-125 says “Intermediate 10 obtained the target compound A09” but should say “Intermediate 10 afforded the target compound A09”.
- In page#6 row#132 says “to gain intermediate 12a-12c.The” but should say “to afford intermediate 12a-12c. The”
- In Scheme 4 legend, aniline is missing for step e.
- In page#7 row#149 says “shown in Tabel 1” but should say “shown in Table 1”.
- In page#7, first time saying ZBG, its meaning should be given in parentheses.
- In page#24 row#535 the degree symbol in “37 C” is not indicated.
Author Response

(The authors gave the same response as above.)

Round 2
Reviewer 2 Report
Re-review of Design, Synthesis and Bioactivity Evaluation of 4,6-disubstituted pyrido[3,2-d]pyrimidine Derivatives as Mnk and HDAC inhibitors by Xing et al.
The corrected manuscript has been substantially improved, as the authors have respond to the majority of the three reviewers’ comments.
Nevertheless, there remain several points to address prior to final acceptance of the ms:
1. The overall ms would benefit from close English language proof-reading, and from correcting some errors like page 1 # 36 "than" is written twice, 2 spaces after "showed" page 3 # 83, and so on….
2. A sufficient discussion of the obtained results is still missing. There is no discussion section contrary to the authors' response.
3. Scheme 1/text: page 4 # 108, the obtained compound is probably A07, and not A02 as mentioned in the text.
Page 4 # 105, when the authors said "Subsequent treatment of the carboxylic acid…" they should specify this treatment is applied to the compound 5.
4. Scheme 3/text: page 6 # 128, o-phenylenediamine is not involved in the step which convert 10 to A09 (step b in the scheme), but in the step d (according to scheme 3). The authors should correct this.
5. Table 2 page 8: to help the lecturer, results obtained with compound A12 should be added to the Table 2.
Author Response
Please see the attachment.
Best regards!
Mr Kun, Xing
